# Reproductive Cycle and Sexual Group Maturity of *Buccinum osagawai* (Neogastropoda: Buccinidae)

**Jung Jun Park** [1], **Hyeon Jin Kim** [2], **So Ryung Shin** [2], **Young Guk Jin** [3], **Jae Won Kim** [4] and **Jung Sick Lee** [2,*]

1   Aquaculture Research Division, National Institute of Fisheries Science, Busan 46083, Korea
2   Department of Aqualife Medicine, Chonnam National University, Yeosu 59626, Korea
3   South Sea Fisheries Research Institute, National Institute of Fisheries Science, Yeosu 59780, Korea
4   Department of Smart Aquaculture, Gangwon State University, Gangneung 25425, Korea
*   Correspondence: ljs@jnu.ac.kr; Tel.: +82-61-659-7172

**Abstract:** This study was performed to obtain information on the basic reproductive ecology and biological resource management of *Buccinum osagawai*. Samples were collected from August 2018 to October 2019 with a drum-shaped net at a depth of 150–250 m along the continental shelf in Jumunjin, on the east coast of Korea. The species showed gonochorism and a lack of external sexual dimorphism. The overall sex ratio (F:M) was 1:1.2 (*n* = 549, F = 250, M = 299); as the size of the individuals increased, the proportion of males tended to decrease. The development pattern of the oocyte was synchronous, in which a dominant group of oocytes was identified within the same ovary. The size of the ripe oocyte was 82.3 (±22.6) × 125.5 (±22.0) μm. The spermatocyte development pattern was group-synchronous, in which multiple stages of germ cell populations were simultaneously identified within the same spermatogenic acinus. The gonad index (GI) for both males and females showed the highest value during June, after which it decreased sharply, and after August it was below 2.0. The main spawning season was from June to July, and the GI and stages of gonadal development did not show a pattern of seasonal changes. The main gonadal active season was from May to July, and both male and female gonadal development and maturation took place over a short period, whereas the recovery period after spawning was longer. At least 60.5% of the group, considering both males and females, showed maturity at 50.1 mm SH or more. Furthermore, the size of 50% group maturity was shown at approximately 50.0 mm SH.

**Keywords:** *Buccinum osagawai*; sex ratio; main spawning period; sexual group maturity

## 1. Introduction

The research on the reproduction of marine organisms provides information that is necessary for understanding their basic biology, as well as management of biological resources and development of aquaculture technology. In such studies, morphological, anatomical, histological, and molecular biological methods all have an important role to play. Among them, the morphological and anatomical methods that analyze sexual maturation and the reproductive cycle based on the size and appearance of the gonads have the advantage of a fast analysis time. Errors may occur during the process of analyzing gonadal development and maturity using histological methods, and errors in the interpretation of the results may occur in all methods. Thus, caution is required with all methods [1,2].

Buccinidae are distributed worldwide in all marine environments, from tropical oceans to the cold waters of the Arctic ocean and Southern ocean, and from the intertidal to the bathypelagic zones. Most demonstrate a preference for a solid seafloor, but some inhabit sandy substrates [3]. Among them, the genus *Buccinum* is mainly distributed in Japan and along the east coast of Korea, and mainly inhabits muddy sands at a depth of 200 to 500 m. However, it is estimated that the habitat extends to about 1000 m, although specimens have only been collected at depths of less than 600 m [4–6].

Within the genus *Buccinum*, studies on *Buccinum undatum* have evaluated the reproductive cycle and seasonal feeding activity [7], the reproductive cycle and energetic cost of reproduction [8], the reproductive cycle and maternal effects on offspring size and number [9], and growth and reproduction [10]. A number of studies on reproduction, including the reproductive cycle and size at sexual maturity [11] of *Buccinum isaotakii*, have been conducted. In these studies, *Buccinum* is gonochoristic, i.e., a gastropod that undergoes internal fertilization [5]. It is known that, even in the same species, reproduction is affected by various environmental factors including water temperature, and consequently may show regional and annual differences [7].

In Korea, 22 genera in the family Buccinidae and 22 species in genus *Buccinum* have been reported [12]. In Korea, *Buccinum* is an expensive edible gastropod mainly captured using drum-shaped nets. Although it is an economically important species, its yield and basic reproductive ecological data have not been reported on. This study investigated sex, the sex ratio, and gonadal development, as well as the main spawning period and sexual group maturity, to provide basic reproductive ecological data and to support the management of biological resources of *Buccinum osagawai*.

## 2. Materials and Methods

### 2.1. Sampling

Samples of *Buccinum osagawai* were collected from August 2018 to October 2019 with a drum-shaped net on the continental shelf at a depth of 150–250 m in Jumunjin, east sea of Korea (Figure 1). A total of 549 specimens (mean shell height 75.2 ± 14.8 mm) were collected (about 35 specimens every month).

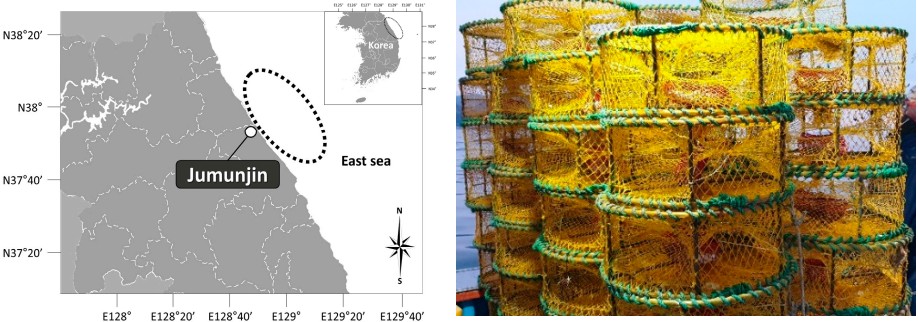

**Figure 1.** Sampling area and drum-shaped net of *Buccinum osagawai*.

### 2.2. Environmental Conditions

Monthly averages of water temperature and salinity profiles in the study area were calculated from daily measurements obtained from the Korea Hydrographic and Oceanographic Administration [13].

### 2.3. Histological Analysis

Specimens were prepared for examination under light microscopy. The hepatopancreas was dissected, which included the gonad; this was fixed in aqueous 10% neutral formalin for 24 h. The fixed sample was rinsed in running water for 48 h, then dehydrated through a graded ethanol series (70–100%), then embedded in paraplast (Leica, Wetzlar, Germany). Embedded tissues were sectioned at a thickness of 4–6 μm using a microtome (RM2235, Leica, Wetzlar, Germany). Samples were stained with Mayer's hematoxylin–0.5% eosin (H–E). Histological quantification of oocytes was performed by converting microscope images into JPEG files, then analyzing these using an image analyzer (i-solution, IMT Inc., New York, NY, USA.) (Figure 2). The ratio of nucleus to cytoplasm of the oocyte was calculated as follows (1).

$$\text{Proportion of nucleus to cytoplasm } (\%) = \frac{\text{Nucleus area } (\mu m^2)}{\text{Ooplasm area } (\mu m^2)} \times 100 \qquad (1)$$

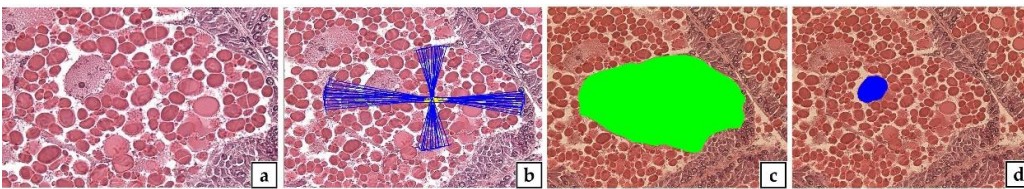

**Figure 2.** Microscopic image analysis of the oocytes. (**a**) original image; (**b**) analysis of oocyte size; (**c**) analysis of cytoplasmic area; (**d**) analysis of nucleus area.

*2.4. Sex Ratio*

The sex ratio (female:male) and percentage of females in the population were computed with the following equations:

$$\text{Sex ratio} = \text{Female } (n)\text{:Male } (n)$$
$$\text{Female } (\%) = [\text{Female } (n)/\text{Female } (n) + \text{male } (n)] \times 100 \tag{2}$$

*2.5. Gonadal Development Stage*

The oocytes were classified into six developmental stages based on size, yolk accumulation, germinal vesicle breakdown (GVBD), and stain ability according to a previous study [14]: oogonium, previtellogenic oocyte, initial vitellogenic oocyte, active vitellogenic oocyte, mature oocyte, and ripe oocyte. The male germ cells were classified into the developmental stages of spermatogonium, spermatocyte, spermatid, and sperm stage. Gonadal development was classified into inactive stage (In), early active stage (Ea), late active and mature stage (Lm), ripe stage (R), and spent and degenerative stage (Sd) for both male and female according to the degree of dominance of each developmental stage of germ cells.

*2.6. Gonad Index (GI)*

The gonad index (GI) was calculated by multiplying each individual by a constant (In = 1, Ea = 2, LM = 3, R = 4, SD = 2) for each gonadal development stage as shown below [15].

$$\text{GI} = \frac{(\text{N of In} \times 1) + (\text{N of Ea} \times 2) + (\text{N of Lm} \times 3) + (\text{N of R} \times 4) + (\text{N of Sd} \times 4)}{\text{Total number}} \times 100 \tag{3}$$

*2.7. Meat Weight Rate (MWR)*

The MWR was calculated using the following equations:

$$\text{MWR} = [\text{meat weight (g)}/\text{total weight (g)}] \times 100 \tag{4}$$

*2.8. Sexual Group Maturity*

Individuals were categorized based on their shell height (SH) into 10 mm intervals. After doing this, for each SH group, we calculated 50% group maturity levels based on the size of the individuals corresponding to the early active stage, late active and mature stage, ripe stage, and spent and degenerative stage of each SH group [16].

*2.9. Statistical Analysis*

Sex ratio data were analyzed using SPSS 18.0 software (SPSS, Inc., Chicago, IL, USA). The observed sex ratio (female:male) for each SH group was compared with the expected ratio of 1:1 using a chi-square test. A *p* value < 0.05 was considered statistically significant.

**3. Results**

*3.1. Sex and Sex Ratio*

The gonad was formed around part of the hepatopancreas, and the mature ovary and testis were white-yellow and yellow, respectively (Figure 3).

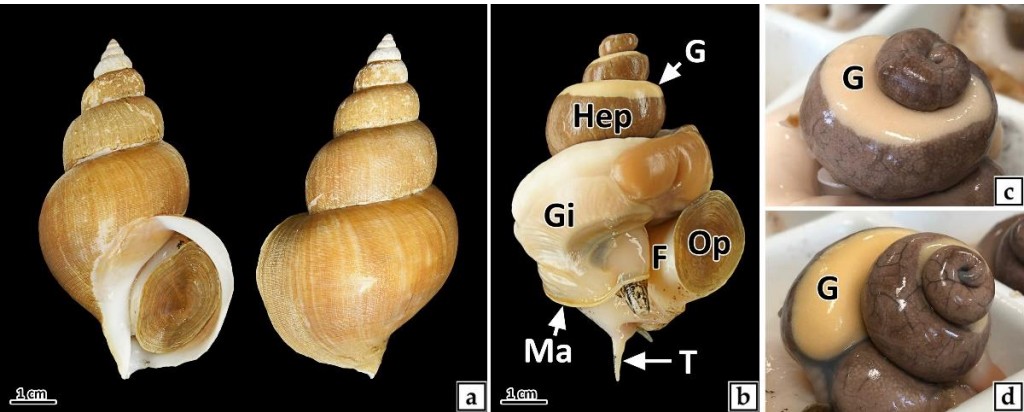

**Figure 3.** Morphology (**a**), anatomy (**b**) and gonad ((**c**): female, (**d**): male) of *Buccinum osagawai*. F: foot; G: gonad; Gi: gill; Hep: hepatopancreas; Ma: mantle; Op: operculum; T: tentacle.

Upon histological analysis of the gonads of all specimens (*n* = 549), synchronous hermaphrodites were not observed, and it was possible to distinguish between males and females in the gonadal inactive stage. The overall sex ratio (F:M) was 1:1.2 (*n* = 549, F = 250, M = 299), with a higher proportion of males. As the size of individuals increased, the proportion of males tended to decrease (Table 1).

**Table 1.** Sex ratio (F:M) with shell height (SH) of *Buccinum osagawai*.

| SH (mm) | Total | Female | Male | Sex Ratio (F %) | Chi-Square | *p* Value |
|---|---|---|---|---|---|---|
| ≤50.0 | 12 | 5 | 7 | 1:1.4 (F 41.7%) | - | - |
| 50.1–60.0 | 73 | 36 | 37 | 1:1.0 (F 49.3%) | 0.014 | 0.907 |
| 60.1–70.0 | 133 | 52 | 81 | 1:1.6 (F 39.1%) | 6.323 | 0.012 |
| 70.1–80.0 | 133 | 57 | 76 | 1:1.3 (F 42.9%) | 2.714 | 0.099 |
| 80.1–90.0 | 101 | 47 | 54 | 1:1.1 (F 46.5%) | 0.485 | 0.486 |
| 90.1–100.0 | 68 | 33 | 35 | 1:1.1 (F 48.5%) | 0.059 | 0.808 |
| 100.1–110.0 | 24 | 16 | 8 | 1:0.5 (F 66.7%) | - | - |
| 110.1≤ | 5 | 4 | 1 | 1:0.3 (F 80.0%) | - | - |
| Total/Mean | 549 | 250 | 299 | 1:1.2 (F 45.5%) | 4.373 | 0.37 |

*3.2. Histological Change of Gonad*

3.2.1. Ovary

The oocyte development pattern was synchronous, in which a dominant group of oocytes was identified within the same ovary. As the oocytes matured, the follicular cells changed from the simple squamous type to the stratified cuboidal and columnar type (Figure 4).

In the ovary of the inactive stage, oogonia and previtellogenic oocytes were the predominant types observed. The shape of the oogonium was circular, the size was 15.5 (±5.3) × 19.5 (±4.4) μm, and the ratio of nucleus to cytoplasm (N/C) was 57.2% (±13.3) (Figure 5). In the karyoplasm of the oogonium, basophilic chromatin was distributed and the cytoplasm was weakly basophilic (Figure 4a). The previtellogenic oocyte was oval shaped, with a size of 31.4 (±5.8) × 41.5 (±8.3) μm. The ratio of N/C was 44.6% (±2.0), showing a decrease from the oocyte stage. The follicle layer was simple and the shape of the follicle cell was squamous (Figure 4b). In the early active stage, initial vitellogenic oocytes were dominant. In the karyoplasm, strongly basophilic nucleolus and chromatin showed a scattered distribution. The cytoplasm was slightly less basophilic and fine yolk granules and small vacuoles were observed (Figure 4c,d). In the late active and mature stage, active vitellogenic oocytes and mature oocytes were dominant. The nucleus of the active vitellogenic oocyte was located near the animal pole. The cytoplasm was filled with large yolk granules that were strongly eosinophilic (Figure 4e,f). The size of the mature oocyte was

60.2 ($\pm$17.4) $\times$ 85.7 ($\pm$21.4) μm (Figure 5). The stainability of the nucleus located near the animal pole became eosinophilic and GVBD (germinal vesicle breakdown) was distinct. The follicle layer was stratified and the shape of the follicle cell was cuboidal and columnar (Figure 4g). The size of the oocytes dominant in the ripe stage was 82.3 ($\pm$22.6) $\times$ 125.5 ($\pm$22.0) μm and the ratio of N/C was 5.6% ($\pm$0.7) (Figure 5). In the spent and degenerative stages, degeneration of the undischarged oocytes within the oogenic acinus was confirmed after spawning. The shape of the degenerative oocytes was irregular and the cytoplasm was weakly basophilic (Figure 4j).

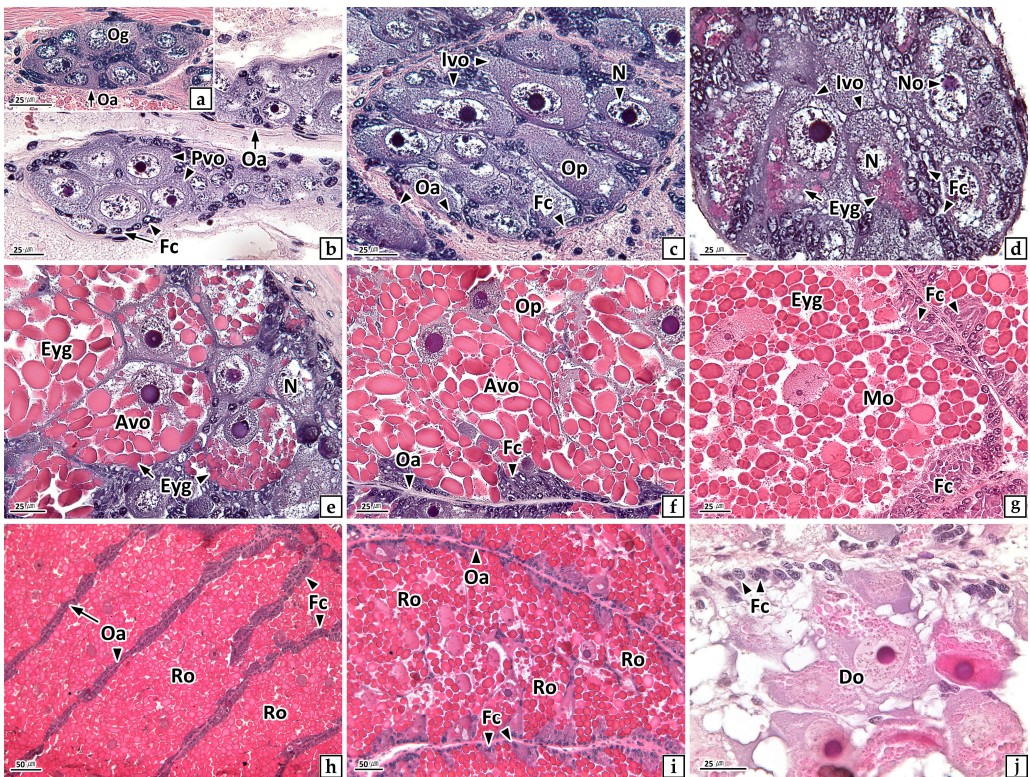

**Figure 4.** Ovarian development stage of *Buccinum osagawai*. H–E stain. (**a**) oogonia (Og) in inactive stage; (**b**) previtellogenic oocytes (Pvo) in inactive stage; (**c**,**d**) initial vitellogenic oocytes (Ivo) in early active stage; (**e**,**f**) active vitellogenic oocytes (Avo) in late active and mature stage; (**g**) mature oocytes (Mo) in late active and mature stage; (**h**) ripe stage; (**i**) ripe oocytes (Ro) in ripe stage; (**j**) spent and degenerative stage. Do: degenerative oocytes; Eyg: eosinophilic yolk granules; Fc: follicle cells; N: nucleus; No: nucleolus; Oa: oogenic acinus; Op: ooplasm.

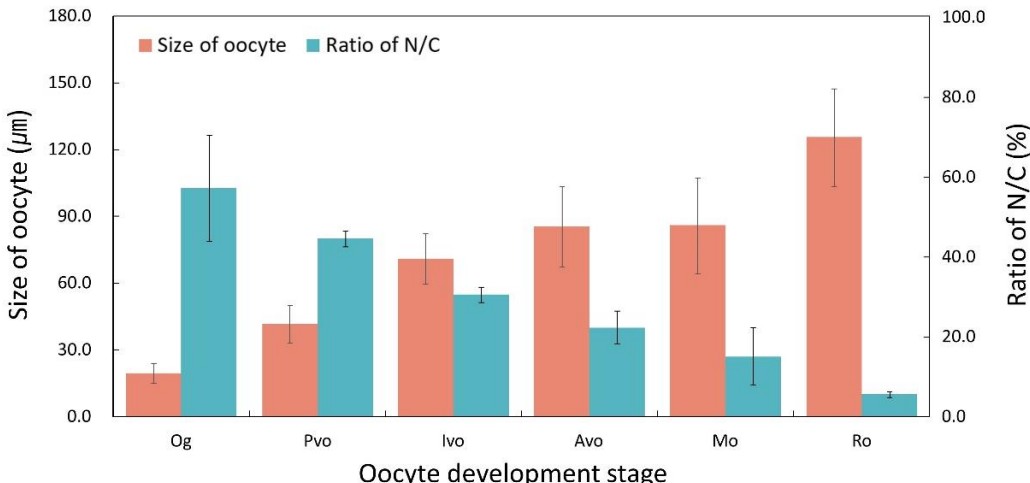

**Figure 5.** Change of the oocyte size and nucleus ratio to cytoplasm during oogenesis in *Buccinum osagawai*. Avo: active vitellogenic oocyte; C: cytoplasm; Ivo: initial vitellogenic oocyte; Mo: mature oocyte; N: nucleus; Og: oogonium; Pvo: previtellogenic oocyte; Ro: ripe oocyte.

### 3.2.2. Testis

The spermatocyte development pattern was group-synchronous, as multiple stages of germ cell populations within the same spermatogenic acinus were identified simultaneously (Figure 6). In the inactive stage of the testis, spermatogonia predominated, and the nucleus occupied most of the cytoplasm (Figure 6a). In the early active stage, spermatocytes predominated, but some spermatogonia and spermatids were also observed (Figure 6b). The spermatocytes had a more condensed nucleolus and cytoplasm than the spermatogonia, and the cytoplasm was weakly eosinophilic upon H–E staining (Figure 6c). In the later active and mature stage, basophilic spermatids predominated in the expanded spermatogenic acinus, though some spermatocytes and sperm were also observed (Figure 6d). At the ripe stage, the spermatogenic acini were filled with strongly basophilic sperm (Figure 6e). In the spent and degenerative stages, residual sperm were observed after the spent (Figure 6f). Finally, during the degeneration process, the spermatogenic acini became distinct, and spermatogonia appeared inside the spermatogenic acini (Figure 6g).

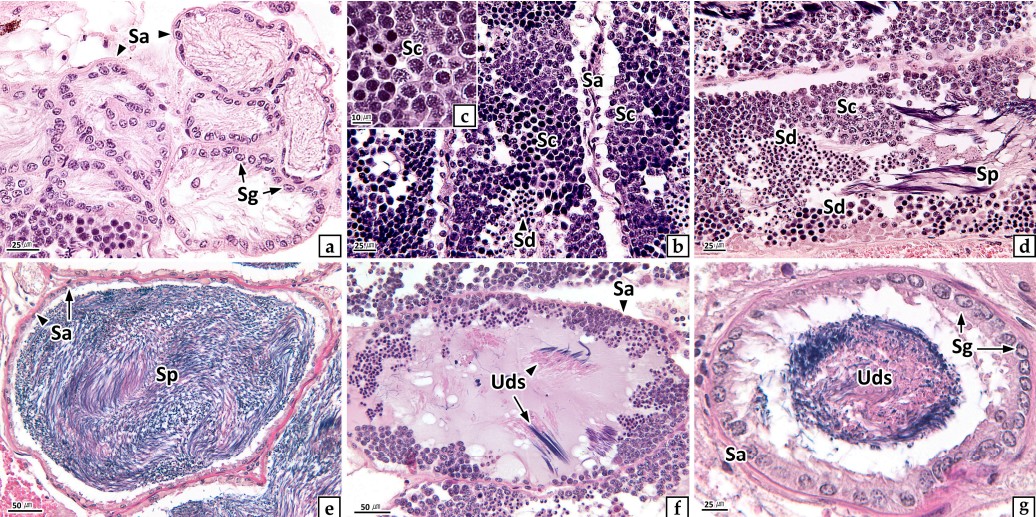

**Figure 6.** Testicular development stage of *Buccinum osagawai*. H–E stain. (**a**) spermatogonia (Sg) in inactive stage; (**b**) early active stage; (**c**) spermatocytes (Sc); (**d**) late active and mature stage; (**e**) sperm (Sp) in ripe stage; (**f,g**) spent and degenerative stage. Sa: spermatogenic acini; Sd: spermatids; Uds: undischarged sperm.

*3.3. Gonadal Development*

3.3.1. Ovary

The annual frequencies (August 2018–July 2019) of the ovarian developmental stages, presented as proportions of the total population, were: inactive stage 5.1%, early active stage 8.3%, late active and mature stage 4.0%, ripe stage 10.7%, and the spent and degenerative stage the highest at 71.9% (Figure 7). Regarding the monthly frequencies of the ovarian developmental stages, the inactive stage dominated during spring, with 44.4% and 16.7% of the population in this stage during April and May, respectively; 58.3% and 28.6% of the population was in the early active stage during May and June, respectively; and 20.8% and 14.3% of the population was in the late active and mature stage during May and June, respectively. Then, 57.1% and 21.4% of the population was in the ripe stage during June and July, respectively, and over 70% of the population was in in the spent and degenerative stage after July (Figure 8).

3.3.2. Testis

The annual frequencies (August 2018–July 2019) of the testicular developmental stages, presented as proportions of the total population, were: inactive stage 7.6%, early active stage 14.3%, late active and mature stage 6.8%, ripe stage 1.9%, and the spent and degenerative stage the highest at 69.4% (Figure 7). Regarding the monthly frequencies of the testis developmental stages, the inactive stage dominated during spring, with 27.8% and 18.8% of the population in this stage during April and May, respectively. Totals of 62.5%, 22.2%, and 70.0% of the population were in the early active stage during May, June, and July, respectively, and 18.8% and 22.2% of the population were in the late active and mature stage during May and June, respectively. A total of 22.2% of the population was in the ripe stage in June, and 33.3% and 20.0% of the population in the spent and degenerative stage during June and July, respectively (Figure 8).

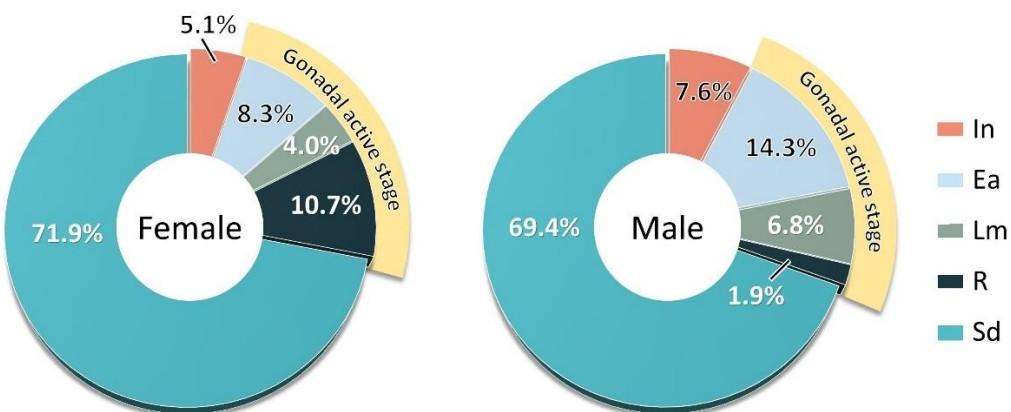

**Figure 7.** Annual frequency (August 2018–July 2019) of gonadal development stage in *Buccinum osagawai*. Ea: early active stage; In: inactive stage; Lm: late active and mature stage; R: ripe stage; Sd: spent and degenerative stage.

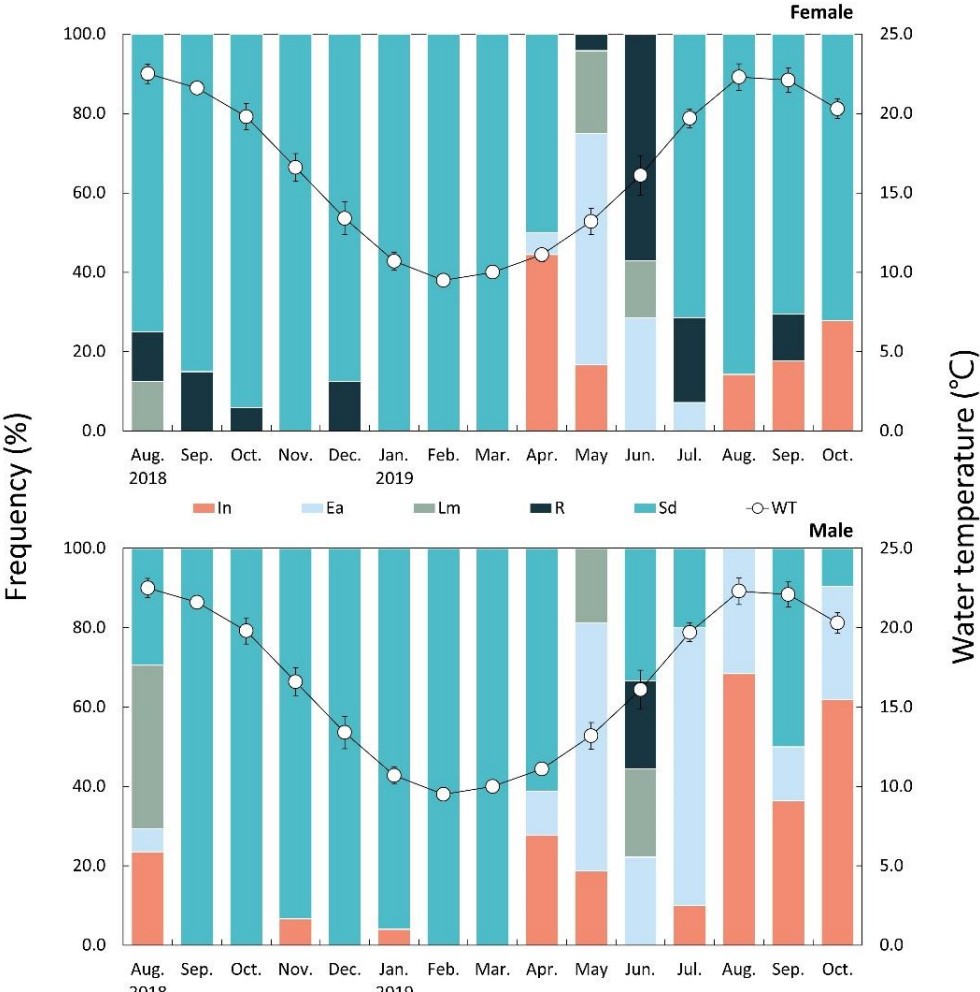

**Figure 8.** Monthly variations of gonadal development stage of *Buccinum osagawai* and water temperature in the sampling area. Ea: early active stage; In: inactive stage; Lm: late active and mature stage; R: ripe stage; Sd: spent and degenerative stage; WT: water temperature. Vertical bars indicate SD.

### 3.4. Environmental Conditions

During the sampling period, the mean water temperature was 16.6 °C and the salinity was 32.8‰. From August 2018 to July 2019, the mean water temperature was 15.4 °C and the salinity was 33.1‰. The water temperature was lowest during February (9.5 °C) and highest during August (22.5 °C in 2018, 22.3 °C in 2019). The water temperature during August, September, and October was similar in both 2018 and 2019. The salinity was higher during winter and lower during summer, but a further pattern was not evident (Figures 8 and 9).

### 3.5. Monthly Change of Gonad Index (GI)

The GI showed a low value of around 2.0 for both males and females until May, increased to the highest values of 3.3 (female) and 2.7 (males) during June, then sharply decreased to a low value below 2.0 after August (Figure 9).

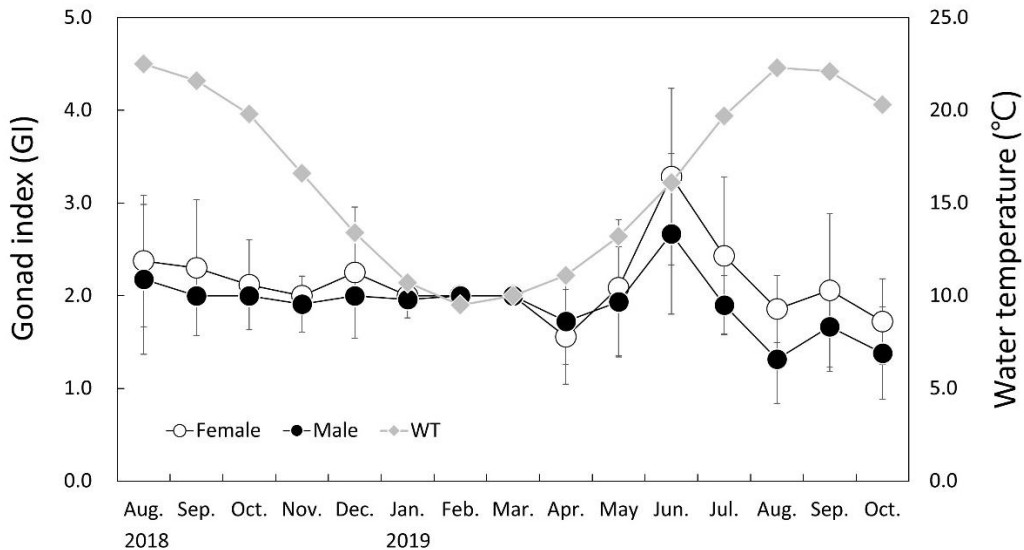

**Figure 9.** Monthly variations of gonad index (GI) in *Buccinum osagawai*. WT: water temperature. Vertical bars indicate SD.

*3.6. Monthly Change of Meat Weight Rate (MWR)*

The mean MWR was 74.1% for females and 71.6% for males. The monthly MWR exhibited the highest values during September 2018 and June 2019 for both males and females, and the lowest values were measured during December 2018; however, there was no seasonal pattern of change (Figure 10).

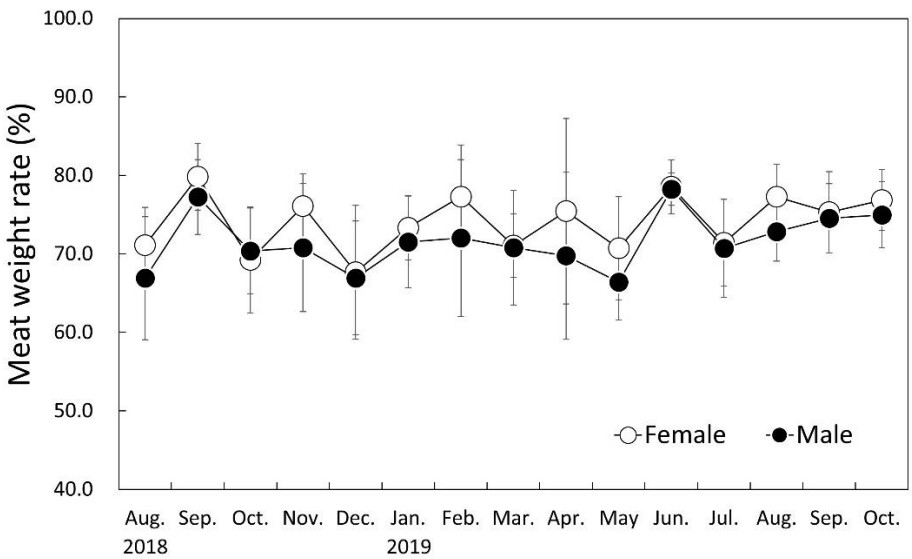

**Figure 10.** Monthly variations of meat weight rate (MWR) in *Buccinum osagawai*. Vertical bars indicate SD.

*3.7. Main Spawning Period*

The main spawning period was analyzed from June to July, which is a period in which the ripe stage and spent and degeneration stage were observed together, and the GI rapidly decreased (Figures 8 and 9).

*3.8. Sexual Group Maturity*

In the group with 50.0 mm SH or less, females (*n* = 5, SH 45.5–49.5 mm) showed 40.0% maturity and males (*n* = 7, 41.6–49.3 mm SH) showed 28.6% maturity. However, in the

group with 50.1–110.0 mm SH, the maturity of females was 73.7–93.8% and that of males was 60.5–78.4% (Table 2).

**Table 2.** Sexual group maturity with shell height (SH) of *Buccinum osagawai*.

| SH (mm) | Female | | | Male | | |
|---|---|---|---|---|---|---|
| | Total | Mature | Maturity (%) | Total | Mature | Maturity (%) |
| ≤50.0 | 5 | 2 | 40.0 | 7 | 2 | 28.6 |
| 50.1–60.0 | 36 | 31 | 86.1 | 37 | 29 | 78.4 |
| 60.1–70.0 | 52 | 46 | 88.5 | 81 | 49 | 60.5 |
| 70.1–80.0 | 57 | 42 | 73.7 | 76 | 54 | 71.1 |
| 80.1–90.0 | 47 | 39 | 83.0 | 54 | 39 | 72.2 |
| 90.1–100.0 | 33 | 26 | 78.8 | 35 | 26 | 74.3 |
| 100.1–110.0 | 16 | 15 | 93.8 | 8 | 6 | 75.0 |
| 110.1≤ | 4 | 4 | 100.0 | 1 | 1 | 100.0 |
| Total/Mean | 250 | 205 | 80.5 | 299 | 206 | 70.0 |

## 4. Discussion

### 4.1. Sex and Sex Ratio

In research of sex in marine animals, sexual dimorphism offers the advantage that morphological determination of sex is possible. Most gastropods do not have sexual dimorphism, but there are species in which males and females can be distinguished by gonadal color at maturity. For example, the gonads of *Batillus cornutus* (Turbinidae) [17], *Haliotis discus hannai* (Haliotidae) [14], and *Haliotis gigantea* [15] are dark green in females and white-yellow in males. For *Neptunea arthritica* (Buccinidae), the testis is reddish brown and the ovary is white-yellow [18]. In *Buccinum osagawai*, external sexual dimorphism was not observed, but mature ovaries exhibited a white-yellow color at maturity, and the testis exhibited a yellow color, making it possible to distinguish between males and females anatomically.

Mollusks can be categorized based on their morphological sex into gonochorism and hermaphroditism. Hermaphroditism is divided into two types: synchronous and asynchronous hermaphroditism [19,20]. Bivalves with asynchronous hermaphroditism undergo sex reversal after spawning [21,22]. Differences in the sex ratios of bivalve populations according to size and age can provide indirect evidence of sex reversal, as has been confirmed in *Mercenaria mercenaria* [21], *Ruditapes philippinarum* [23], *Tegillarca granosa* [24], and *Mytilus coruscus* [25].

Through histological analysis of the gonads, synchronous hermaphroditism was not observed in *Buccinum osagawai* as in other gastropods [15,26–28], and it was confirmed that their sexual system is gonochoristic since it was possible to distinguish between males and females even at the gonadal inactive stage.

In populations of marine animals, the sex ratio is one of the important pieces of information for prediction of population size change and has implications for growth differences between sexes. The sex ratio within marine gastropods varies according to size group, even for the same taxon and habitat. In general, sex ratios in herbivorous gastropods show a strong female bias [29].

The sex ratio (M:F) of *Turbo torquatus* (Turbinidae) collected from Wollongong in New South Wales, Australia in one study was 1:1.25, showing a significant difference, but there was no significant difference in those collected from Ulladulla (206 males, 200 females) and Eden (168 males, 193 females). In another study, the overall sex ratio (F:M) of *Amphissa acutecostata* (Columbellidae), a deep-sea gastropod widely distributed in the Northeast Atlantic, was about 0.6:1 (63 females, 94 males); there were 10 males and 2 females at a station depth of 940 m and 28 males and 15 females at a station depth of 1283 m. In another study, the overall sex ratio of *Gymnobela subaraneosa* (Raphitomidae) was measured at 32 males and 26 females, but at a station depth of 1500 m was 1 female to 9 males [30]. Furthermore, the sex ratio of *Buccinanops globulosus* (Nassariidae) from the Golfo Nuevo of Argentina showed a generally higher number of females [31].

In Muricidae, the sex ratio (M:F) of *Hexaplex trunculus* from the tidal areas in the Gulf of Gabès of southern Tunisia has been measured at 1:1.6, showing a significant difference, and as individual size increased, the proportion of females tended to increase [26]. The overall sex ratio (F:M) of *Plicopurpura pansa* in one study was about 0.9:1, with the male proportion significantly higher and a tendency toward a 1:1 ratio as size increased [32]. The overall sex ratio (M:F) of *Bolinus brandaris* has been measured at 1:1.5, with the proportion of females higher; this sex ratio showed seasonal differences. In addition, with shell lengths of 5 mm or less, males were a higher proportion, but as size increased, the proportion of females tended to increase [33]. In contrast, the sex ratio (M:F) of *Rapana venosa* from the Black sea has been measured at about 1:1 in a two-year-old group, but 1.3:1, 3.3:1, 4:1, 17.7:1 in the three-, four-, five-, and six-year-old groups, respectively, and in all groups older than six years all individuals were male [34]. In another study, the sex ratio (F:M) of *Rapana venosa* over 55 mm in length from the Trabzon coast of Turkey in the southeastern Black sea was 15% female and 85% male. However, there was no significant difference among those from Samsun, Turkey (46.3% females and 53.7% males) [35].

In Buccinidae, the sex ratio (F:M) of *Buccinum undatum* has been measured at 1:0.88, with a high proportion of females [8]. The sex ratio (M:F) of *Babylonia areolata* was measured at 1:1.3 in one study in those with shell height of 35–50 mm, and 1:2.6 in the largest group, with a high female proportion [36].

In this study, the overall sex ratio (F:M) of *Buccinum osagawai* was 1:1.2, with a high proportion of males. As with many gastropods, the proportion of females tended to increase as individual size increased. The tendency for the proportion of females to increase as the individual size increases is related to growth rate and life expectancy; further research is needed on this topic. An increase in females is considered to be a reproductive strategy advantageous to maintaining and expanding population size, but the phenomenon of an increase in the proportion of males requires further interpretation in relation to environmental conditions of their habitats. In addition, analysis of the same number of individuals per size group (age) is necessary to fully determine the sex ratio in a population.

*4.2. Gonadal Development and Maturity*

The gonadal structure of gastropods differs according to taxon, and histological information of the ovary is very important for interpretation of the oocyte development pattern and yolk accumulation mechanism. In Vetigastropoda, the structure of the ovaries of *Batillus cornutus* [17], *Haliotis discus hannai* [14], and *Haliotis gigantea* [15] shows numerous oogenic follicles similar to those of bivalves. These oocytes, as in those in bivalves, possess egg stalks, which are transport channels for vitellogenin [37,38].

However, many gastropods, including *Bolinus brandaris* [39] and *Buccinum undatum* [10], have an acinus type ovary. Their oocytes do not possess an egg stalk (Table 3), but instead the oocytes are surrounded by follicle cells involved in the vitellogenin pathway and oocyte resorption after spawning [40,41]. The ovary of *Buccinum osagawai* was of the acinus type as in other Buccinidae, and follicle cells tended to develop along with oocytes. In gastropod oogenesis, the difference in egg size according to presence or absence of an egg stalk, i.e., an exovitellogenous pathway, is not clear, and currently not much information exists regarding the difference in egg size; thus, it is necessary to gather data on this.

**Table 3.** The presence or absence of egg stalk and egg size in some gastropods.

| Order | Family/Species | Egg Stalk | Egg Size (μm) | Citation |
|---|---|---|---|---|
| Vetigastropoda | Haliotidae | | | |
| | *Haliotis discus hannai* | + | 202.9 × 142.1 | [14] |
| | *Haliotis gigantea* | + | 200 | [15] |
| | Turbinidae | | | |
| | *Batillus cornutus* | + | 140 | [17] |
| Neogatropoda | Buccinanopsidae | | | |
| | *Buccinanops cochlidium* | - | 220 | [27] |
| | Buccinidae | | | |
| | *Buccinum isaotakii* | - | 240 ± 40 | [11,42] |
| | *Buccinum osagawai* | - | 82.3 × 125.5 | Present study |
| | Busyconidae | | | |
| | *Busycotypus canaliculatus* | - | - | [28] |
| | Columbellidae | | | |
| | *Amphissa acutecostata* | - | 99.06 | [30] |
| | Nassariidae | | | |
| | *Nassarius festivus* | - | 160 | [43] |
| | Terebridae | | | |
| | *Hastula cinerea* | - | 300 × 150 × 100 | [44,45] |
| | Raphitomidae | | | |
| | *Gymnobela subaraneosa* | - | 114.82 | [30] |
| | Muricidae | | | |
| | *Bolinus brandaris* | - | - | [39] |
| | *Thais carinifera* | - | 200–220 | [46,47] |
| | Volutidae | | | |
| | *Adelomelon beckii* | - | 222 | [48] |
| | *Adelomelon brasiliana* | - | 180 ± 8 | [49,50] |
| | *Odontocymbiola magellanica* | - | - | [51] |
| | *Zidona dufresnei* | - | - | [52] |

Oocyte development patterns provide information on spawning and reproductive patterns [53]. Wallace and Selman [54] classified the modes of oocyte development into synchronous, group-synchronous, and asynchronous patterns according to the distribution of dominant oocytes in the ovary of teleost. The synchronous pattern is one in which all oocytes are formed, grown, and spawned simultaneously. The group-synchronous pattern is one in which at least two populations of oocytes are distinct at the same time. The asynchronous pattern is one in which oocytes of all stages are present, without dominant populations.

This classification of oocyte developmental pattern is applicable to gastropods as well. *Adelomelon beckii* (Volutidae) displays a group-synchronous pattern and has two spawning seasons per year [41]. *Turbo torquatus* displays a synchronous pattern, and spawns twice per year [55]. *Batillus cornutus* [17], *Haliotis discus hannai* [14], and *Haliotis gigantea* [15] display the group-synchronous pattern, and these spawn multiple times during one spawning season per year. *Buccinum undatum* shows a synchronous pattern of spawning once per year [9,10], while *Buccinum osagawai* shows a similar synchronous pattern and seems to spawn once per year.

The reproductive cycle and gonadal development are affected by various environmental factors such as water temperature, salinity, food, and photoperiod. Among these, the influence of water temperature is greatest for neritic species, but for deep-sea species, other factors such as food are more important than water temperature [53,56].

Changes in gonad index (GI) and gonadal developmental stage of Turbo torquatus [55], *Adelomelon brasilianum* [49], *Adelomelon beckii* [41], *Batillus cornutus* [17], *Ceratostoma rorifluum* [57], *Hexaplex trunculus* [26], *Monetaria annulus* [58], *Buccinum undatum* [10], *Bolinus brandaris* [39], *Haliotis discus hannai* [14], and *Haliotis gigantea* [15], which mainly inhabit the neritic zone, show seasonal rhythms in relation to water temperature. However, the sexual

patterns of the deep-sea gastropods *Amphissa acutecostata* and *Gymnobela subaraneosa* do not show seasonal rhythms [30].

Changes in GI and gonadal development of *Buccinum osagawai* also did not show a seasonal rhythm. This is attributed to their main habitat being an aphotic pelagic region, not a neritic zone, and thus the water temperature change is relatively low at a depth of about 200 m.

During the reproductive cycle in gastropods, the gonadal activity period differs depending on the species, but is relatively long. The reproductive period of *Nassarius festivus* is from September to May of the following year [43], while *Hexaplex trunculus* [26], *Buccinanops cochlidium* [27], and *Bolinus brandaris* [39] have a long gonadal activity period and a relatively short resting phase. The gonadal mature stage of *Monetaria annulus* is more than six months [58]. The gonadal maturation of *Buccinum undatum* takes place over a relatively long period, from February to October [10].

In contrast, the main gonadal active period of *Buccinum osagawai* was short (May–July), and the annual frequency of the gonadal active stage was about 23%. These results indicate that gonadal development and maturation in both males and females takes place over a short period, and the recovery period after the spent stage is long. However, further research is needed to discover the cause of these reproductive differences between *Buccinum osagawai* and other species.

Information on sexual group maturity is very important for understanding the reproductive ecology and for the management of biological resources. At least 66.7% of the *Buccinum osagawai* population, in both males and females, showed maturity at a shell height of 50.1 mm or more. Although the sample number of specimens with less than 50.0 mm shell height was small, at least 50% of the group showed maturity at approximately 50.0 mm.

## 5. Conclusions

*Buccinum osagawai* is gonochoristic and lacks external sexual dimorphism. Mature ovary and testis were white-yellow and yellow in color, respectively. The overall sex ratio (F:M) was 1:1.2 ($n$ = 537, F = 246, M = 291), with a higher proportion of males; however, as the size of individuals increased, the proportion of males tended to decrease. Future studies on the growth rate according to age are needed. The main spawning season was from June to July, and the GI and stages of gonadal development did not show a pattern of seasonal changes. Both males and females showed at least 60.5% maturity at 50.1 mm SH or more. The size of 50% group maturity was shown at about 50.0 mm SH. We conclude that fishing individuals under the size at which 50% group maturity is reached (50.0 mm SH) and during the main spawning season (June-July) should be prohibited to maintain the fishery resources of this species. The reproductive index did not show a pattern of seasonal changes, and further investigation is required into the factors affecting maturation.

**Author Contributions:** Conceptualization, J.J.P., H.J.K. and J.S.L.; methodology, H.J.K., S.R.S. and J.S.L.; software, Y.G.J. and J.W.K.; validation, H.J.K. and J.S.L.; formal analysis, J.J.P., H.J.K. and J.S.L.; investigation, J.J.P., H.J.K., S.R.S. and J.W.K.; resources, Y.G.J. and J.W.K.; data curation, J.J.P., H.J.K. and J.S.L.; writing—original draft preparation, J.J.P. and J.S.L.; writing—review and editing, H.J.K. and J.S.L.; visualization, H.J.K., Y.G.J., J.W.K. and J.S.L.; supervision, J.J.P. and J.S.L.; project administration, J.S.L.; funding acquisition, J.J.P., Y.G.J. and J.S.L. All authors have read and agreed to the published version of the manuscript.

**Funding:** This research received no external funding.

**Institutional Review Board Statement:** Not applicable.

**Data Availability Statement:** Not applicable.

**Acknowledgments:** This work was supported by a grant from the National Institute of Fisheries Science, South Korea (R2022004).

**Conflicts of Interest:** The authors declare no conflict of interest.

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
