# Peer review of "Reproductive Cycle and Sexual Group Maturity of Buccinum osagawai (Neogastropoda: Buccinidae)"

_fishes, doi:10.3390/fishes7050267_

Round 1
Reviewer 1 Report
Fishes-190-772
Gonadal Structure, Sex Ratio, Main Spawning Period and Sexual Group Maturity of Buccinum osagawai (Neogastropoda: Buccinidae)
The present study aimed to find the basic information on reproductive characteristics of Buccinum osagawai caught form east coast of Korea, in Jumunjin. No external dimorphism was found to distinguish the sexes. The sex ratio (F:M) was 1:1.2.The females showed synchronous development type of oocytes and the size of oocyte during the final maturation was 82-125 um. In males, multiple stages of germ cell populations were simultaneously identified within the same spermatogenic acinus. The highest values of GSI were found in June to July for both sexes. More than 50% of matured Buccinum osagawai was in 50 mm shell height size.
The manuscript can be accepted, but there are some corrections to improve the manuscript before publication.
Some of my comments are listed below:
Title:
The title of the MS is too long and could be shorten.
Abstract
Line 11: it should be 82.3-125.5 um, if size variation. If not, please indicate this is dimension.
Keywords: All the keywords repeated in title and should be change accordingly.
Introduction
Page 2, line 4: change occur to distribute.
Page 2, line 16-19: give appropriate and related citations
Materials and methods:
Page 2: how many times were the samples taken?
Results:
Page 4: It seems to prepare the section environmental conditions into the materials and methods.
Is it possible to prepare information of month or season variation of oocyte sizes as graphs?
Page 7, section 3.4.2: change it as un-bold text.
Page 8, section 3.5: please provide statistical differences throughout the year for gonad index.
Page 9, section 3.6: please provide statistical differences throughout the year for meat weight rate.
Discussion:
Well writing, but just focus on your results.
Why sex ratio was change when the gastropod gets higher size? It should be noted in aspects of some physiological features in the discussion.
Page 11: Table 3 is not necessary and can be omitted from discussion.
Page 12: The authors said that "The reproductive cycle and gonadal development are affected by various environmental factors such as water temperature, salinity, and food". What about the other main factor generally "photoperiod"?
Page 12, line 20-23: change the names as italic.
Conclusions:
It is very similar to results and abstract! It should be rewritten.
Reviewer 2 Report
The manuscript entitled “Gonadal Structure, Sex Ratio, Main Spawning Period and Sexual Group Maturity of Buccinum osagawai(Neogastropoda: Buccinidae)” by Jung Jun Park , Hyeon Jin Kim , So Ryung Shin , Young Guk Jin , Jae Won Kim , Jung Sick Lee, was described the important information of reproduction in Buccinum osagawai, However, some of the conclusions need to be further confirmed, and the writing problems also need to be improved.
Major concerns,
1. Throughout, the English language needs a native speaker to improve.
2. A detailed description of life history, distribution, and phylogenetic relationships would make the introduction more useful than a discussion of the advantages and disadvantages of morphology and histology. I suggest the introduction needs to be reorganized more logically.
3. Results section, the describe order of the description should be the same as the order in which the diagram appears, otherwise it will cause confusion for the reader. Subheadings can be divided according to morphological observations, histological observations, etc., to make the description more logical.
4. Figure 4 and Figure 6, the Oogonia and Spermatogonia needs biomarkers to be further confirmed, especially spermatogonia.
Minor concerns,
1. The results section 3.1 the salinity data "32.8%" and "33.1%" would be ‰.
2. There need statistics of significance in Figure 5;
3. 3.1 Environmental Conditions, this section should be remove to M&M section.
4. The manuscript lack of serious number of line, this make review difficult.
Reviewer 3 Report
GENERAL COMMENTS
This manuscript gives information on the basic reproductive ecology of Buccinum osagawai off the east coast of Korea.
The most important aspects are related to the fact that there is no information about the species, although being commercial important. Histological analysis was performed in a considerable sample, providing good information to be analysed and great histological images.
Statistical analysis was not performed for any result obtained, and this needs to be performed, even to obtain a solid discussion and eventually see patterns that were not observed.
Although one of the goals was “to support management of biological resources of Buccinum osagawai”, nothing is specifically suggested or said on how the results presented could improve this.
Wish the authors all the best.
SPECIFIC COMMENTS
ABSTRACT
- Can be reduced to the main results in simple sentences. Think about the reader and what are the main achievements that deserve to be in this section.
INTRODUCTION
- Detail the economic importance of the species: landings evolution, prices, how it’s the resource used, and what are the impacts.
MATERIAL AND METHODS
Gonadal Development Stage
- Lacks reference to the maturity stage used.
Gonad index
- Lacks reference to the formula used.
Sexual Group Maturity
- Lacks reference to the criteria used.
Statistical Analysis
- No statistical analysis was performed to compare variations throughout the year. Parametric and non-parametric analysis can and should be used.
RESULTS
- Environmental Conditions
- Could be important to mention in the abstract. Nothing is said about it.
Sex and Sex Ratio
- Perform the Chi-square test in a sample with less than 50 individuals it’s not correct. There are several SH classes where it should have not been performed and the results could be incorrect.
Ovary and testes
- The long paragraphs describing the maturity stages could be transformed into tables easier to read.
- Figure 5 - the two graphs can be merged into one.
Gonadal development
- Analyse possible differences between months using statistics. Differences would be a highlight of the abstract and can be discussed in more detail.
Monthly Change of Gonad Index (GI)
- Can also be statistically analysed.
- Figure 9 – “Total” series don’t seem necessary.
Sexual Group Maturity
- Is there any specific model to estimate size at first maturity in mollusks as it has in fish? From a quick reading, it seems there is.
DISCUSSION
- There are some long sentences, very descriptive that can be summed. Please simplify the reading.
- Should incorporate the recommendations made throughout the manuscript.
Reviewer 4 Report
Reviewer comments
The article has well written and has more interesting observation on reproductive biology of Buccinum osagawi from Korean coast. This manuscript can be accepted after these minor corrections.
Abstract
· Replace the word “The sex is gonochorism”
· More focus is given on the histological analysis of gonad. It can be reduced
· Write a two-line of conclusion of the study in the abstract itself
Introduction
· Include a paragraph on Buccinum osagawi about basic biological data and its importance in environment and fisheries
· Whether only B. undatum and B. isaotakki has been studied? cross-check it
Materials and Methods
· Drum-shaped net? A picture of those net in operation while collecting samples can be added
· Rephrase “For specimen preparation for light microscopy”
· Hepatopancreas has gonad? “hepatopancreas was dissected, which included the gonad”
· Figure 2: Can be more clear and the toolbox in the figure should be removed
· Gonadal development stage – Is there a standard reference or author’s modified it. Pls clarify
· Gonad Index (GI) – Give reference for the index
Results
· Is there a correlation between water temp and Gonadal stages ?
· Is there only a single spawning period ? or there is a secondary spawning period
Discussion
· Discussion can have sub-heading as in case of results
· Discussion on sex-ratio can be reduced and should be focussed on the Genus Buccinum
· It is better to compare the results with the Neogastropod. Rather than species from other orders
Conclusion
· It seems to the repetition of the abstract. It is written very casually. Conclusion should have the importance of the study and way forward.
